# A Convolutional Neural Network Classifier Identifies Tree Species in Mixed-Conifer Forest from Hyperspectral Imagery

Geoffrey A. Fricker [1,2,*], Jonathan D. Ventura [3], Jeffrey A. Wolf [4], Malcolm P. North [5], Frank W. Davis [6] and Janet Franklin [2]

1 Department of Social Sciences, California Polytechnic State University, San Luis Obispo, CA 93407, USA
2 Department of Botany and Plant Sciences, University of California, Riverside, CA 92521, USA; jfrankl@ucr.edu
3 Department of Computer Science and Software Engineering, California Polytechnic State University, San Luis Obispo, CA 93407, USA; jventu09@calpoly.edu
4 Amazon Web Services, Amazon Corp., Seattle, WA 98109, USA; jawolf@amazon.com
5 U.S. Forest Service, PSW Research Station, Mammoth Lakes, CA 93546, USA; mnorth@ucdavis.edu
6 Bren School of Environmental Science & Management, University of California, Santa Barbara, CA 93106, USA; fwd@bren.ucsb.edu
* Correspondence: africker@calpoly.edu; Tel.: +1-805-756-1150

**Abstract:** In this study, we automate tree species classification and mapping using field-based training data, high spatial resolution airborne hyperspectral imagery, and a convolutional neural network classifier (CNN). We tested our methods by identifying seven dominant trees species as well as dead standing trees in a mixed-conifer forest in the Southern Sierra Nevada Mountains, CA (USA) using training, validation, and testing datasets composed of spatially-explicit transects and plots sampled across a single strip of imaging spectroscopy. We also used a three-band 'Red-Green-Blue' pseudo true-color subset of the hyperspectral imagery strip to test the classification accuracy of a CNN model without the additional non-visible spectral data provided in the hyperspectral imagery. Our classifier is pixel-based rather than object based, although we use three-dimensional structural information from airborne Light Detection and Ranging (LiDAR) to identify trees (points > 5 m above the ground) and the classifier was applied to image pixels that were thus identified as tree crowns. By training a CNN classifier using field data and hyperspectral imagery, we were able to accurately identify tree species and predict their distribution, as well as the distribution of tree mortality, across the landscape. Using a window size of 15 pixels and eight hidden convolutional layers, a CNN model classified the correct species of 713 individual trees from hyperspectral imagery with an average F-score of 0.87 and F-scores ranging from 0.67–0.95 depending on species. The CNN classification model performance increased from a combined F-score of 0.64 for the Red-Green-Blue model to a combined F-score of 0.87 for the hyperspectral model. The hyperspectral CNN model captures the species composition changes across ~700 meters (1935 to 2630 m) of elevation from a lower-elevation mixed oak conifer forest to a higher-elevation fir-dominated coniferous forest. High resolution tree species maps can support forest ecosystem monitoring and management, and identifying dead trees aids landscape assessment of forest mortality resulting from drought, insects and pathogens. We publicly provide our code to apply deep learning classifiers to tree species identification from geospatial imagery and field training data.

**Keywords:** deep learning; species distribution modeling; convolutional neural networks; hyperspectral imagery

## 1. Introduction

### 1.1. Background and Problem

Automated species mapping of forest trees using remote sensing data has long been a goal of remote sensing and forest ecologists [1–11]. Conducting remote inventories of forest species composition using an imaging platform instead of field surveys would save time, money, and support analysis of species composition over vast spatial extents [12–15]. Accurate assessments of tree species composition in forest environments would be an asset for forest ecologists, land managers, and commercial harvesters and could be used to study biodiversity patterns, estimate timber stocks, or improve estimates of forest fire risk. Operational remote sensing and field sensors are improving, but new classification tools are necessary to bridge the gap between data-rich remote sensing imagery and the need for high-resolution information about forests. Our work sought to improve current automated tree species mapping techniques.

Tree species mapping from remote sensing imagery has proven a difficult challenge in the past [11], owing to the lack of (1) widely-available high resolution spatial and spectral imagery; (2) machine learning classifiers sophisticated enough to account for the lighting, shape, size, and pattern of trees as well as the spectral mixing in the canopies themselves; and (3) spatially precise ground data for training the classifiers. Efforts to overcome these challenges have taken different approaches with regard to remote sensing data sources and classification techniques. High-resolution multispectral satellite remote sensing [16], hyperspectral airborne imagery [2,17], and even airborne Light Detection and Ranging (LiDAR, see Table A1) without spectral imagery [18] have been used to discriminate tree species. Many methods take a data fusion approach, combining LiDAR with multispectral [19,20] or hyperspectral imagery [21–23] to classify tree species. To discriminate individual tree crowns at both high spatial and spectral resolution, airborne imagery is required. Hyperion Imaging Spectrometer was the first and only imaging spectrometer to collect science-grade data from space [24] and it has been used to map minerals [25], coral reefs [26], and invasive plant distributions from orbit [27]. The 30 m spatial resolution and low signal to noise ratio make it an unsuitable instrument for mapping individual trees; however, spaceborne hyperspectral sensors have demonstrated viability to map areas inaccessible to airborne platforms such as the Airborne Visible/Infrared Imaging Spectrometer (AVIRIS) [28].

The use of airborne hyperspectral imagers in forested environments was advanced by the Carnegie Airborne Observatory (CAO) used for large swaths of carbon rich forests in the Airborne Spectronomics Project, mapping canopy chemistry, functional plant traits, and individual tree species in diverse tropical forests [29–32]. The CAO uses hyperspectral imagery combined with LiDAR allowing for a three-dimensional, chemical characterization of the landscape based on spectral absorption features and is informative about the composition of plants communities. This sensor combination was adopted by the National Ecological Observation Network (NEON) Airborne Observation Platform (AOP), providing openly available data at 81 monitoring sites in 20 eco-climatic domains across the conterminous USA, Alaska, Hawaii, and Puerto Rico [33]. AOP imaging instruments include a small-footprint waveform LiDAR to measure three-dimensional (3D) canopy structure, a high-resolution hyperspectral imaging spectrometer, and a broadband visible/shortwave infrared imaging spectrometer. Data are collected at a spatial resolution (sub-meter to meter) sufficient to study individual organisms and observe stands of trees. In addition to manned aerial flights, unmanned aerial vehicles have recently been used to identify tree species using hyperspectral imagery and point cloud data [34–36].

Over the past decade, there have been significant advances in the application of a variety of machine learning classifiers to hyperspectral imagery for tree species classification. Classifiers have included Random Forest, a decision tree method, Support Vector Machines, and artificial neural networks, and have been applied to (sub)tropical wet and dry forests [4,37], temperate and boreal forests [8,9,38], plantations and agroforestry [10,39,40], and urban forests [41–43]. These machine learning classifiers achieved accuracies (averaged across species) ranging from 63%–98% when applied

to 4–40 tree species using tens to occasionally hundreds of trees per species for training. Classification accuracies typically varied more widely among species in these studies (e.g., per-species accuracies from 44%–100%) than among machine learning and other classifiers when they were compared.

Convolutional Neural Networks (CNNs) are machine learning supervised classifiers that, in addition to characterizing spectral signatures, analyze the spatial context of the pixel. To our knowledge, CNNs have not been applied to tree species classification from airborne hyperspectral imagery. CNNs can perform concurrent analysis of spectra and shape using multiple deep layers of pattern abstraction which are learned through numerical optimization over training data. In this study, we parameterized and tested a CNN classifier applied to high-resolution airborne hyperspectral imagery of a forested area for tree species identification with sparsely distributed training labels.

### 1.2. Convolutional Neural Networks

Convolutional Neural Networks (CNNs) give high performance on a variety of image classification and computer vision problems. CNNs use computational models that are composed of multiple convolved layers to learn representation of data with multiple levels of abstraction [44,45]. These algorithms have dramatically improved the state of the art in image recognition, visual object recognition, and semantic segmentation by discovering intricate structure in large data sets. CNNs consist of many sets of convolution and pooling layers separated by non-linear activation functions (such as the rectified linear unit [ReLU]). These "deep learning" models are trained using the backpropagation algorithm, and variants of stochastic gradient descent [45]. CNNs have been used for over two decades in applications that include handwritten character classification [46], document recognition [47], traffic sign recognition [48], sentence classification [49] and facial recognition [50]. Biological imaging applications of CNNs include identifying plant species based on photographs of leaves [51], interpreting wildlife camera trap imagery [52], and new crowdsourced applications such as the iNaturalist mobile application, which uses user photographs, geographic locations and CNNs to identify species of plants and animals [53].

Application of CNN methods has been limited by computational resources, and the need to program the code to apply the neural network and backpropagation algorithms to a classification problem from scratch [54]. Over the past decade, graphical processor units (GPUs) that support massive parallelization of matrix computations, GPU-based matrix algebra libraries, and high-level neural network libraries with automatic differentiation have made training and application of neural networks accessible outside of research labs in industry settings.

CNNs are conceptually well-suited to spatial prediction problems. Geospatial images contain spatially structured information. In the case of trees, the spatial structure of canopies is related to tree size relative to pixel size; in high resolution imagery if a pixel falls in a tree canopy its neighbors are also likely to be in the same canopy and have similar information [55]. Information in neighboring pixels is related to information in a focal pixel and these relationships decay with distance. CNN classifiers operate on this same principle to find patterns in groups of nearby pixels and relate them to 'background' information. For automated species mapping, individual trees are represented as clusters of similar pixels at fine spatial resolution and, at a coarser spatial scale, stands of trees are clusters of individuals, both of which might be informative in determining the species of each tree.

CNNs have been applied to moderate spatial and/or spectral resolution imagery to classify land use [56] and reconstruct missing data in Moderate Resolution Imaging Spectrometer (MODIS) and Landsat Enhanced Thematic Mapper remotely sensed imagery [57]. Deep Recurrent Neural Networks have also been used to classify hyperspectral imagery with sequential data, such as spectral bands [58]. CNNs have been used to classify high spatial resolution imagery into land use and land cover categories [59–63], but not tree species.

### 1.3. Research Objectives

Our work seeks to evaluate the use of CNNs for automated tree species mapping from airborne hyperspectral data. We implemented and evaluated a Deep CNN supervised classifier applied to airborne hyperspectral high-resolution imagery to discriminate seven tree species, as well as dead trees, in temperate mixed-conifer forest in western North America. Imagery was acquired by the NEON AOP in a region of the Southern Sierra Nevada, California, USA, at 1-m spatial resolution. Field data collected for this study were used to train the classifier, while independently collected data were used for testing. We had the following objectives:

1. Evaluate the application of CNNs to identify tree species in hyperspectral imagery compared to a Red-Green-Blue (RGB) subset of the hyperspectral imagery. We expected improved ability to accurately classify trees species using hyperspectral versus RGB imagery.
2. Assess the accuracy of the tree species classification using a test dataset which is distinct from the training and validation data.
3. Demonstrate potential uses of high-resolution tree species maps, i.e., analyze the distribution of trees across an elevation gradient.
4. Provide tools so that other geospatial scientists can apply such techniques more broadly and evaluate the computational challenges to upscaling to larger areas.

## 2. Materials and Methods

### 2.1. Study Site and Airborne Imaging Data

AOP data were collected in July 2017 across NEON's D17 Pacific Southwest study region. The airborne remote sensing LiDAR and hyperspectral imagery data were collected from June 28th to July 6th in 2017 by the NEON AOP onboard a DeHavilland DHC-6 Twin Otter aircraft. The hyperspectral data is collected using a pushbroom collection instrument (AVIRIS next-gen) that was designed and built by the National Aeronautics and Space Administration's Jet Propulsion Laboratory. It measures reflected light energy in 426 spectral bands extending from 280 to 2510 nm with a spectral sampling interval of five nm. Data were collected at approximately 1000 m above ground level, with a cross-track Field of View of 34 degrees and an instantaneous field of view of 1.0 milliradian, resulting in a ground resolution of approximately 1 m [33,64]. NEON performed the atmospheric correction of the NEON Imaging Spectrometer using ATCOR4r and then orthorectified and output the imagery onto a fixed, uniform spatial grid using nearest neighbor resampling.

For this study, we used a south-to-north oriented strip of hyperspectral imagery data approximately 16 km long by 1 km wide covering a portion of the Teakettle Experimental Forest (TEF) and extending up slope towards the alpine zone (Figure 1). The TEF is in the Southern Sierra Nevada Mountains (36°58′00″, 119°01′00″), approximately 80 km east of Fresno, CA. Owing to differences in solar illumination across imagery strips collected at different times and dates by the AOP, we chose to focus only on a single strip of imagery to test the CNN classifier. The imagery strip was selected because it includes forest plots established by North et al. [65] in the TEF as part of a long-term ecological field experiment, allowing us to use self-collected, detailed, georeferenced data on tree species identity, combined with tree inventory data from the TEF long-term experiment, to test the CNN classifier (Figure 1).

TEF supports four main forest types. Mixed conifer comprises about 65% of the forested area, predominantly between 1900 and 2300 m elevation. Jeffrey pine (*Pinus jeffreyi*) (5.5%) is prevalent on shallow soil conditions within the mixed-conifer type. Red fir (*Abies magnifica*) (28%) dominates elevations > 2300 m except for very moist locations where lodgepole pine (*Pinus contorta*) (0.5%) occurs in almost pure stands. Within the mixed-conifer forest, we found a fine-scale mosaic of four patch types: closed forest canopy, shrub patches dominated by mountain whitethorn (*Ceanothus cordulatus*), open gaps, and areas of rock and extremely shallow soils. In contrast, red fir forests are

more homogenous with greater, more continuous canopy cover and higher tree basal area and density than mixed conifer [65]. The imagery strip or transect covers an elevation gradient spanning 1935 m to 2630 m, encompassing a transition in forest composition zone. At lower elevations, some broadleaf oak trees with dense shrub understory are found, and pine and cedar conifer species are abundant. In the middle elevations, ~2200 m, there are large stands of lodgepole pine that surround flat wetlands and seasonal drainages. At the higher elevations (>2500 m), red and white fir (*Abies concolor*) (conifers) dominate, with fewer pine and cedars and no oak trees. At the highest elevations, there are several large rock outcrops and steep terrain, with no road access, so there were no field data collected in the upper ~20% of the imagery strip (Figure 1).

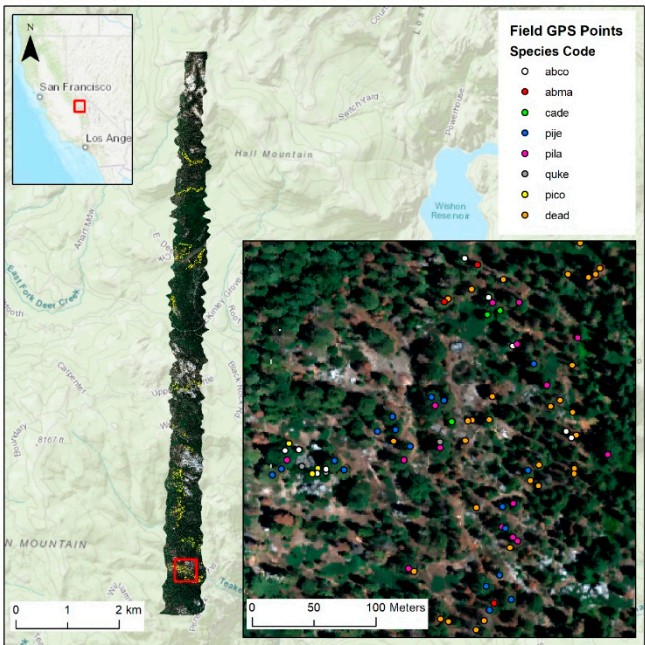

**Figure 1.** National Ecological Observation Network (NEON) Airborne Observation Platform Red-Green-Blue hyperspectral subset orthophoto imagery showing the location of the strip of hyperspectral imagery used in analysis (left) intersecting the Teakettle Experimental Forest (TEF) Watershed north-east of Fresno, CA (upper left inset). Zoomed view at the southern end of the flight transect (area in red square) and an example of individual tree points collected in the field using high precision Global Positioning System (GPS), colored based on the species (seven species) or mortality status (lower right). Field GPS Point training data collected by the authors and by University of New Mexico field teams in the TEF permanent plots are shown as yellow and are located within transects crossing the flightline (left).

## 2.2. Field Data and Study Species

In September 2017, two months after image acquisition, we collected field-based observations of individual tree crowns along eight transects oriented roughly east-west across the imagery strip at different elevations to train, test, and validate the CNN model (Figure 1). Systematic sampling throughout the imaged area allowed us to collect a large sample of tree crown locations for all common tree species in this mixed-conifer forest. The seven species of trees and one class of dead trees (any species) used in our study are shown in Table 1 with the number trees from each class. Hereafter, we refer to the species by their common names in text and use the four-letter abbreviation in figures.

**Table 1.** The scientific name, common name, four-letter abbreviated species code, and the number of trees per species used in our study.

| Code | Scientific Name (Common Name) | Abbreviation | Number |
|:---:|:---:|:---:|:---:|
| 0 | *Abies concolor* (White fir) | abco | 119 |
| 1 | *Abies magnifica* (Red fir) | abma | 47 |
| 2 | *Calocedrus decurrens* (Incense cedar) | cade | 66 |
| 3 | *Pinus jeffreyi* (Jeffrey pine) | pije | 164 |
| 4 | *Pinus lambertiana* (Sugar pine) | pila | 68 |
| 5 | *Quercus kelloggii* (Black oak) | quke | 18 |
| 6 | *Pinus contorta* (Lodgepole pine) | pico | 62 |
| 7 | Dead (any species) | dead | 169 |

We used high-precision Global Positioning System (GPS) to measure the locations of trees across the eight transects and recorded species, stem diameter at breast height (1.4 m) ("DBH"), and mortality status (dead or alive). These transects were oriented along both sides of unpaved logging and access roads to maintain relatively easy walking across the swath of the flight-line. We selected isolated, large-diameter "overstory" trees with large canopies, as well as smaller trees with canopies that were isolated from surrounding canopies, so that these trees could be identified in the LiDAR derived Canopy Height Model (CHM) and differentiated from the surrounding trees. Our objective was to associate the field measured species identity with the associated pixels in the hyperspectral imagery (to minimize misidentification owing to lack of perfect registration between GPS and imagery) and minimize mixing of species (resulting from overlapping crowns) in the training data. Our sampling prioritized coverage of the elevation gradient and imagery extent. We aimed to sample about 100 trees of each species and a range of tree sizes (>5 m tall). We also sampled standing dead trees of all species but ultimately grouped them in the same prediction class (dead tree), because often the tree species cannot be positively identified a few years after it dies. We aimed to sample each tree species in all locations throughout the image strip, but some species like black oak were very rare as isolated overstory individuals except at lower elevations, while at the highest elevation red and white fir dominate.

To record the tree locations, we used the mobile ESRI Collector Application on Android and IOS devices and two EOS Arrow Gold Differential GPS receivers/antennas (Terrebonne, Quebec, Canada) with SBAS real-time satellite correction to achieve sub-meter spatial GPS precision. Points that exceeded 1 m in horizontal precision were not used. We recorded GPS locations on the southern side of each tree and from a location about two thirds of the distance from the stem to the edge of the crown whenever possible, for consistency and better satellite visibility, but there were instances where local conditions required us to take a GPS measurement on a different side of the stem. A total of 920 tree crown canopies were originally recorded in the field, but 619 were eventually used in our final analysis because of a selective filtering process. The filter process removed trees with unacceptable horizontal GPS accuracy (>1 m), canopies that were too close to each other and could not be reliably discriminated from neighboring canopies, and trees with incomplete canopy coverage on the margins of the imagery strip.

The problem of accurate field positioning and identifying large individual trees in the field that can be referenced to remote sensing imagery has proved challenging. Global Positioning System (GPS) signal is broadcast in the microwave L-bands which are well known to scatter amongst tall, dense tree canopies making them particularly good for estimating forest biomass [66,67] and particularly bad for GPS signal acquisition [68]. An accurate correspondence needs to be established between field and remote sensing measured individual trees, and difficulties arise when individual trees are mapped on the ground related to GPS uncertainties in closed canopy conditions [69]. In this study we attempt to reduce these uncertainties by using an additional L5 band (in addition to L1 and L2 GPS bands), and by only focusing on isolated trees reducing ambiguity between field GPS measurements and LiDAR tree locations.

In addition to our field transects which spanned the flight line strip at different elevations, we augmented our data with other tree map data that were not part of the same field sampling effort [70,71]. We utilized existing detailed stem maps for 9 of 18 4-ha permanent plots collected by University of New Mexico field teams over the summer of 2017 in the TEF as part of an experimental study of fire and thinning effects on forest structure and dynamics [65,72]. The plots were established in 1998. Data collected for each tree (>5 cm DBH) in a plot include location in geographic space and plot coordinate space, species, DBH, and condition. Experimental treatments include burning, overstory thinning, understory thinning, and control plots with three replicates of each combination. The most recent (2017) Teakettle re-survey censused the burned tree plots, improved spatial accuracy of tree locations relative to earlier GPS surveys (which were therefore not used), and was conducted concurrently with NEON AOP imagery collection. We combined the 2017 Teakettle tree census, conducted for the nine burned plots intersecting the flight line, with the transect data, to train, test and validate the CNN classifiers. Our dataset comprised 713 trees (Table 1), including 94 from Teakettle plots and 619 from transects.

### 2.3. Label Data Preparation for CNN Classification

To prepare the imagery labels for CNN classification, we used the LiDAR derived canopy height model (CHM) to manually digitize individual tree canopies from the TEF experimental plots and the transect data so that pixels within those canopies could receive species labels (Figure 2). GPS points were aligned with the 1-meter spatial resolution LiDAR CHM and 'pseudo true-color' (Red-Green-Blue; RGB) subset of the hyperspectral imagery. We used a subset of the bands which were closest to the band centers for red (band 55, 0.64–0.67 microns), green (band 37, 0.53–0.59 microns), and blue (band 20, 0.45–0.51 microns). In some cases, small displacements (1–3 m) between the LiDAR CHM and imagery were detected, and in these cases, care was taken to label the canopy area in the imagery and not the canopy crown in the CHM.

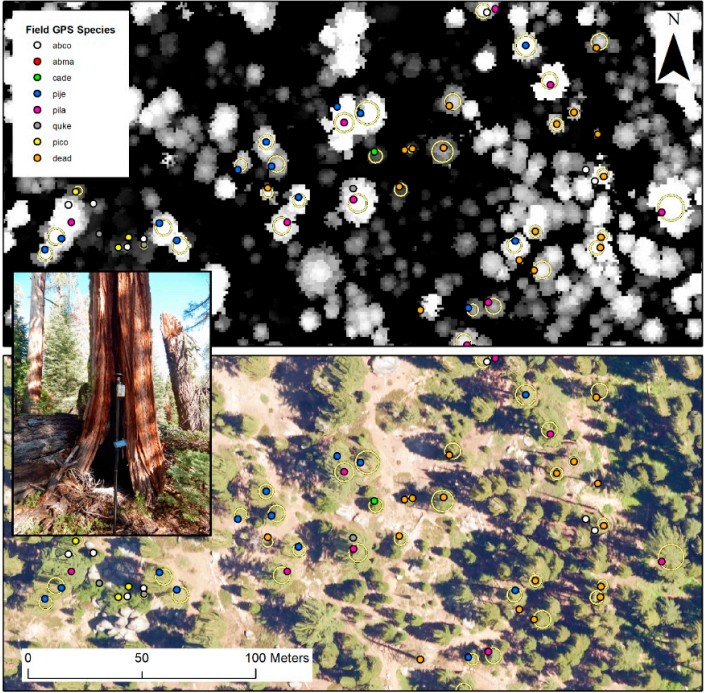

**Figure 2.** High precision GPS points, colored by species on the Light Detection and Ranging (LiDAR) derived Canopy Height Model (top), the high resolution ortho photos with points labeled using the field measured diameter at breast height (DBH) in centimeters (bottom). Digitized canopy label outlines are shown as yellow circles. The GPS antenna taking a static position next to an Incense cedar (*cade*) tree in the field (center inset).

We created circle polygons centered on the center of each tree of mapped stems in high resolution aerial imagery. We labeled the polygons with the class of the species and rasterized these polygons. Each circle was digitized to include 'pure pixels' of only canopy and avoid background, soil, or other canopies and as a result the circles vary in size and orientation relative to the original high-precision GPS point. These tree polygons were considered individual units of the analysis and the pixels within each polygon were treated as the same class, so that individual pixels in the same polygon are not considered independent by the model. The rasters were of the same spatial resolution and same coordinate reference system (CRS) as the hyperspectral imagery, i.e., there was a 1:1 mapping of pixels in the label raster and pixels in the hyperspectral image. The label raster includes a no-data value for each pixel where there was no tree polygon and an integer encoding of the class where there was a polygon. Labeled pixels were extracted from the hyperspectral data and used when training the CNN (Figure 2).

To rigorously evaluate our method, we prepared a k-fold cross validation experiment. We divided the tree polygons into 10 folds and trained the model 10 times, each time holding out one of the folds as testing data and using the remaining folds for model training. To ensure that each fold had similar spatial coverage, we first applied k-means clustering to the polygon centroids with k = 10 to group the trees into 10 clusters corresponding roughly to the field transects. We then split each cluster into 10 folds, so that the *i*-th fold for the entire dataset consists of the *i*-th fold from each cluster.

## 2.4. CNN Model Architecture

We evaluated a fully convolutional neural network architecture [73]. We trained the model on $L \times L \times D$ patches from the training data, where D was a constant value of 32 for the hyperspectral image (after applying dimensionality reduction by principal components analysis (PCA)) or three for the RGB image and *L*, the patch size was a hyperparameter. The network architecture consisted of a cascade of $3 \times 3$ convolutional layers followed by a final $1 \times 1$ convolutional layer for the output. The number of output channels at each layer starts at C for the first layer and doubles at each subsequent layer up to a maximum number of channels C', where C and C' are hyperparameters. The final layer has eight output channels corresponding to the eight classes in our dataset. We applied L2 regularization with a strength of $\alpha$, where $\alpha$ is a hyperparameter. We did not use zero padding, and the number of convolutional layers was chosen as $(L-1)/2$, so that the CNN would output a single prediction for the center pixel in the window under consideration. The patch size affects how much of the image the network can see when making a prediction. The number of filter kernels affects the discriminative power of the network, more filter kernels means a more powerful network but might lead to overfitting to the training data. The regularization strength affects how strongly the network weights are dampened, dampening the weights helps combat overfitting. All layers except the final layer used Rectified Linear Units (ReLU) activations as their non-linear activation function. The final activation layer was a softmax function that outputs the probability of each class. We then applied an argmax function to obtain the class label for the class with the highest probability. We implemented the models in Keras using the TensorFlow backend (Appendix B).

At test time, the model is run in a fully convolutional manner to produce a prediction at each output pixel in parallel. We did not use max pooling layers so that the output size of the network would be the same as the input except for a $(L-1)/2$ border of missing predictions at the edges of the input. At test time we divide the raster into overlapping tiles and run the network on each tile in order to fit the computation into memory; the overlap between tiles accounts for the missing predictions at the borders of the tiles. A schematic of the model architecture and data processing workflow can be found in Figure 3 below.

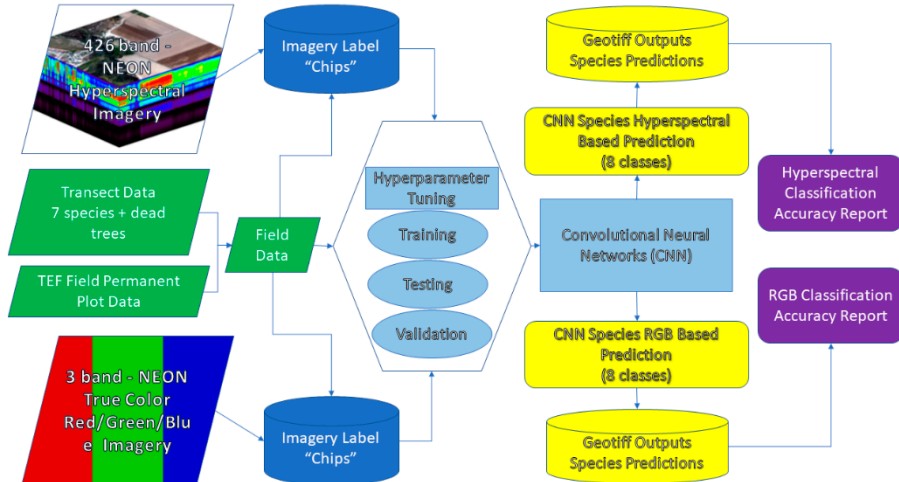

**Figure 3.** Schematic of the data processing flow and architecture. Datasets (left), Field data (green), convolutional neural network (CNN) model (light blue), CNN Predictions (yellow) Accuracy Reports (violet).

## 2.5. Optimization/Hyperparameter Tuning/Prediction/Assessment

To assemble the training data we extract an $L \times L \times D$ patch from the input raster centered at each labeled pixel inside tree polygons from the training set. During training we randomly withhold 10% of the patches as validation data which is used to monitor for overfitting and aid in hyperparameter selection. The data splits were the same for the hyperspectral and RGB data models. In each experiment, the tree polygons in the training set are distinct from those in the test set, so that no pixels from tree polygons in the test set are present in the training set. The training subset is used for optimization of model parameters relative to a loss function. The validation subset is used for selecting a single model from the set of models with different hyperparameters. The validation set is necessary because the hyperparameters are not learned by the optimization process. We calculate and report the precision, recall and F-scores for each species and for all species combined for each split for both sets of imagery.

We manually searched for the best hyperparameter settings by training multiple models on the training dataset and comparing resulting accuracy on the validation dataset. We applied stochastic gradient descent with momentum to optimize each model, with a learning rate of 0.0001, and momentum of 0.9. We used categorical cross-entropy as a loss function. A batch size of 32 was used for each training iteration. For data augmentation we rotated the patches in 90-degree increments and also flipped the original and rotated patches horizontally and vertically. We trained each model over 20 epochs, or 40 epochs for the RGB model, retaining the model with the best validation accuracy during the training process. After choosing a single model from the collection of trained models based on maximum validation set accuracy, we evaluated the accuracy of this model on the test set. Raster labels used for training, testing and validation were derived from both the field data from our field transects and plot data from the TEF permanent plots. The accuracy is measured based on the majority of predicted pixels in each polygon, each of which is assigned to one species.

In order to address objectives 1 and 2, prediction accuracy is reported using Precision, Recall, and the F-Score. Precision is the proportion of the image correctly classified when compared to test data, describes the ability of a classification model to return only relevant instances, and is often called "user's accuracy," and recall is the proportion of the true observations correctly classified, describes the ability of a classification model to identify all relevant instances, and is also called "producer's accuracy" in the remotely sensed literature [74]. The F-measure is the harmonic mean of precision and recall. The classified image was used to calculate the relative proportion of each class in elevation belts as a demonstration of an ecological application of the classifier (objective 3). Relative proportion was

used to normalize for total tree cover variation in the image and emphasize the relative abundance of species, a commonly-used measure in forest ecology [75].

## 3. Results

### 3.1. CNN Classification and Parameter Settings

The best hyperparameter settings were a patch size of $L = 15$, regularization strength of $\alpha = 0.001$, and $C = 32$ filter kernels in the first convolutional layer up to a maximum of $C' = 128$ kernels. We also found it beneficial to use a balanced loss function because the proportion of examples of each class in the dataset is imbalanced. We weighted the categorical cross entropy loss function with class weights that were inversely proportional to the number of examples of each class in the training set.

Using Tensorflow and Keras, we applied our final CNN classifiers to all tree crown pixels in both the RGB true-color and hyperspectral imagery strip by excluding pixels where the CHM height was below a threshold of 5 m. Each tree polygon was assigned a classification based on the species (or mortality) prediction of the majority of pixels in each polygon, which was compared to the species recorded in the field. The CNN classifier using the hyperspectral imagery outperforms the RGB subset image whether measured by precision, recall or F-Score, and across all species/mortality status. In the hyperspectral and RGB CNN models the precision, recall and F-score were 0.87 and 0.64, respectively. The combined F-Scores for the hyperspectral CNN model were 0.87 for the five cross-validated testing subsets (Table 2). Confusion matrices for all classes based on the hyperspectral CNN and RGB models are reported in Tables 3 and 4 respectively and predictions for the hyperspectral imagery are shown in Figure 4.

For the hyperspectral model (Table 2), at the genus level, pines (*Pinus*) had higher precision, recall, and F-scores than firs (*Abies*), while incense cedar (*Calocedrus*) had intermediate recall, precision and F-score, and oaks (*Quercus*), with a relatively small sample size, had low recall (0.61) and precision (0.73). At the species level, the F-scores from high to low were Jeffrey pine (0.95), sugar pine/lodgepole pine (0.93), incense cedar (0.88), white fir (0.78), red fir (0.74), and black oak (0.67). Dead trees slightly outperformed the average F-score (0.88).

The RGB model performed poorly compared to the hyperspectral model at the class level across all metrics (precision, recall, F-Score; Table 2). The relative discrimination of genus level differences was qualitatively similar to the hyperspectral results. Pines (*Pinus*) generally outperformed firs (*Abies*). Oaks (*Quercus*) performed only slightly worse in the RGB model compared with the hyperspectral model (F-scores of 0.65 and 0.67 respectively) and cedars (*Calocedrus*) performed poorly overall (0.47 F-score). At the species level, the F-scores from high to low were Jeffrey pine (0.69), sugar pine (0.67), black oak (0.65), lodgepole pine (0.50), white fir (0.49), incense cedar (0.47), and red fir (0.35). Dead trees far outperformed the average F-score (0.87) (Table 2).

**Table 2.** Summary results table reporting the precision, recall and F-Score for the average of 10 folds of the k-fold cross validation for both the Red-Green-Blue subset and hyperspectral imagery.

| Species | Species Code | Hyperspectral | | | RGB | | |
|---|---|---|---|---|---|---|---|
| | | Precision | Recall | F-score | Precision | Recall | F-Score |
| White fir | 0 | 0.76 | 0.81 | 0.78 | 0.46 | 0.53 | 0.49 |
| Red fir | 1 | 0.76 | 0.72 | 0.74 | 0.41 | 0.30 | 0.35 |
| Incense cedar | 2 | 0.90 | 0.85 | 0.88 | 0.50 | 0.44 | 0.47 |
| Jeffrey pine | 3 | 0.93 | 0.96 | 0.95 | 0.65 | 0.73 | 0.69 |
| Sugar pine | 4 | 0.90 | 0.96 | 0.93 | 0.67 | 0.68 | 0.67 |
| Black oak | 5 | 0.73 | 0.61 | 0.67 | 0.69 | 0.61 | 0.65 |
| Lodgepole pine | 6 | 0.84 | 0.87 | 0.86 | 0.54 | 0.47 | 0.50 |
| Dead | 7 | 0.90 | 0.85 | 0.88 | 0.88 | 0.86 | 0.87 |
| Ave/Total | | 0.87 | 0.87 | 0.87 | 0.64 | 0.64 | 0.64 |

The greatest per-species misclassification rate based on the hyperspectral classification was for black oak, where six of the 17 trees in the sample were classified as a lodgepole pine and one as dead (Table 3). Red and white firs were often confused with each other and rarely other species. There were 11/47 red fir classified as white fir, 7/119 white fir classified as red fir and nine white fir incorrectly classified as dead. Jeffrey pine and sugar pine had the highest F-scores and misclassifications were evenly distributed across the other species classes. Incense cedar was most often misclassified as white fir and Jeffrey pine (three occurrences each) or dead (four occurrences). Dead trees were most often misclassified as white fir (14 occurrences), red fir (four occurrences), Jeffrey pine (three occurrences), and lodgepole and sugar pine (two occurrences each) (Table 3).

For the RGB model, the greatest per-species misclassification rate was for red fir (0.35 F-score), where 16/47 of the trees in the sample were classified as a white fir, while only 14 trees were classified correctly (Table 4). Red and white firs were often confused with each other and often with Jeffrey pine, incense cedar, and dead trees. Jeffrey pine and sugar pine had the highest F-scores and misclassifications were mostly between the two pine species, although 16 Jeffrey pine trees were misclassified as white fir. Incense cedar most often misclassified as white fir (15 occurrences) and Jeffrey pine (nine occurrences) or lodgepole pine (six occurrences). Dead trees were most often misclassified as white fir (nine occurrences), Jeffrey pine (six occurrences), and incense cedar (four occurrences each), although the RGB classification had the most success correctly classifying dead trees (Table 4). Mapped predictions for the RGB model are not shown because they are less accurate compared to the hyperspectral model and of low enough accuracy to not be of practical use because the proportion of samples is imbalanced.

**Table 3.** The hyperspectral classification confusion matrix for tree species identified using high-precision field GPS (rows) compared to the CNN model species prediction (columns; numbered labels refer to species numbers shown in row labels).

| Species | Abbrev | Spp Code | 0 | 1 | 2 | 3 | 4 | 5 | 6 | 7 | Recall | F-Score |
|---|---|---|---|---|---|---|---|---|---|---|---|---|
| 0. White fir | abma | 0 | 96 | 7 | 2 | 2 | 2 | 0 | 1 | 9 | 0.81 | 0.78 |
| 1. Red fir | abco | 1 | 11 | 34 | 1 | 0 | 0 | 0 | 0 | 1 | 0.72 | 0.74 |
| 2. Incense Cedar | cade | 2 | 3 | 0 | 56 | 3 | 0 | 0 | 0 | 4 | 0.85 | 0.88 |
| 3. Jeffrey pine | pije | 3 | 1 | 0 | 2 | 158 | 1 | 0 | 1 | 1 | 0.96 | 0.95 |
| 4. Sugar pine | pila | 4 | 1 | 0 | 1 | 1 | 65 | 0 | 0 | 0 | 0.96 | 0.93 |
| 5. Black oak | quke | 5 | 0 | 0 | 0 | 0 | 0 | 11 | 6 | 1 | 0.61 | 0.67 |
| 6. Lodgepole pine | pico | 6 | 0 | 0 | 0 | 2 | 2 | 4 | 54 | 0 | 0.87 | 0.86 |
| 7. Dead | dead | 7 | 14 | 4 | 0 | 3 | 2 | 0 | 2 | 144 | 0.85 | 0.88 |
| | | Precision | 0.76 | 0.76 | 0.90 | 0.93 | 0.90 | 0.73 | 0.84 | 0.90 | | |

**Table 4.** The Red-Green-Blue classification confusion matrix for tree species identified using high-precision field GPS (rows) compared to the CNN model species prediction (columns; numbered labels refer to species numbers shown in row labels). Polygon (per-tree) from five combined k-folds cross validation test datasets.

| Species | Abbrev | Spp Code | 0 | 1 | 2 | 3 | 4 | 5 | 6 | 7 | Recall |
|---|---|---|---|---|---|---|---|---|---|---|---|
| 0. White fir | abma | 0 | 63 | 11 | 9 | 17 | 3 | 0 | 5 | 11 | 0.53 |
| 1. Red fir | abco | 1 | 16 | 14 | 3 | 9 | 1 | 0 | 2 | 2 | 0.30 |
| 2. Incense Cedar | cade | 2 | 15 | 2 | 29 | 9 | 3 | 1 | 6 | 1 | 0.44 |
| 3. Jeffrey pine | pije | 3 | 16 | 3 | 5 | 119 | 12 | 1 | 5 | 3 | 0.73 |
| 4. Sugar pine | pila | 4 | 3 | 1 | 4 | 12 | 46 | 0 | 2 | 0 | 0.68 |
| 5. Black oak | quke | 5 | 0 | 0 | 0 | 1 | 1 | 11 | 4 | 1 | 0.61 |
| 6. Lodgepole pine | pico | 6 | 14 | 1 | 4 | 9 | 2 | 2 | 29 | 1 | 0.47 |
| 7. Dead | dead | 7 | 9 | 2 | 4 | 6 | 1 | 1 | 1 | 145 | 0.86 |
| | | Precision | 0.46 | 0.41 | 0.50 | 0.65 | 0.67 | 0.69 | 0.54 | 0.88 | |

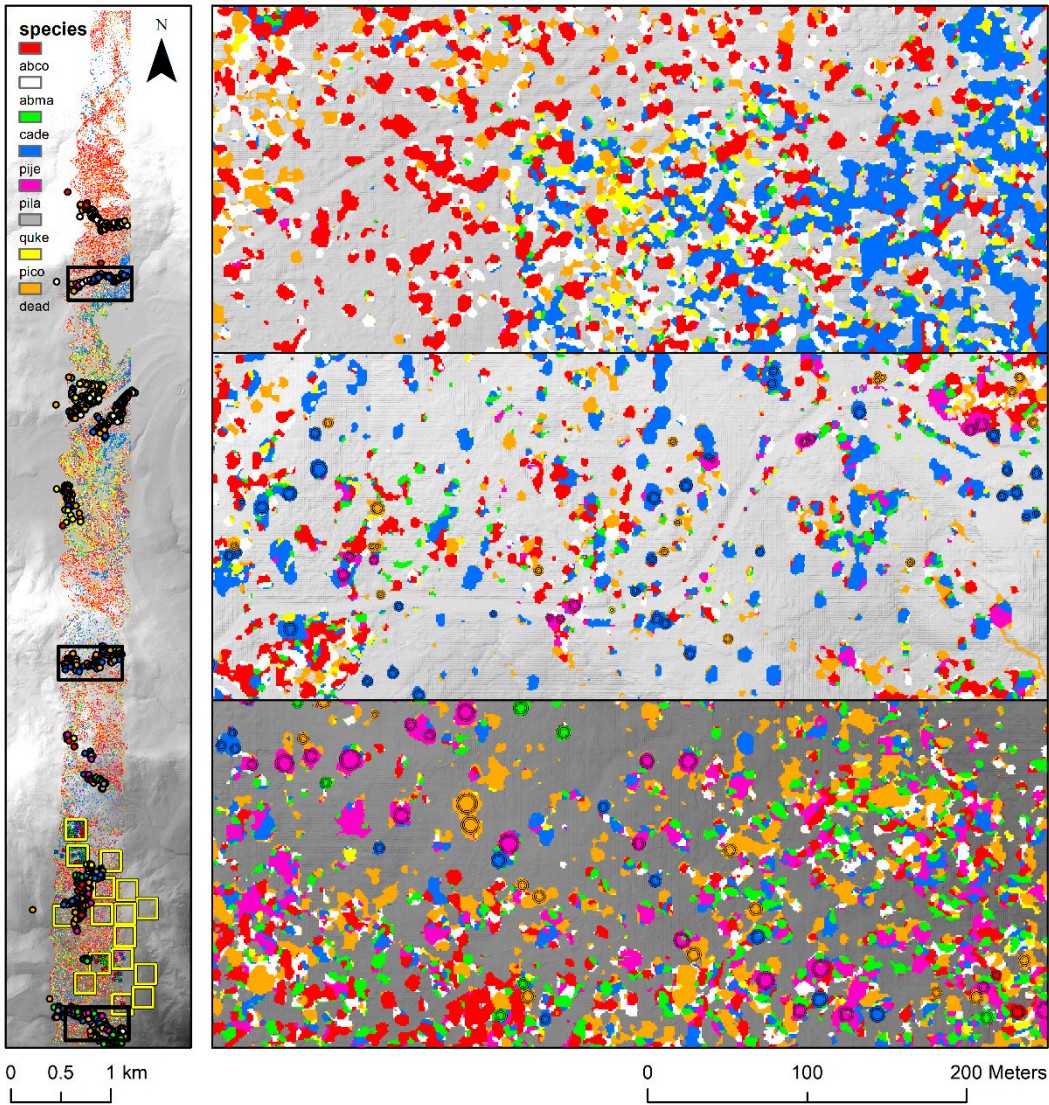

**Figure 4.** Full flight line strip showing hyperspectral CNN predicted species as colors on a hill shade digital elevation model (left), Field GPS points are represented as points (left) and circular polygons (right) and CNN model prediction results are represented as the colorized raster image for three sections at high, mid, and lower elevation sites.

### 3.2. Application of High-Resolution Tree Species Mapping

The high-resolution tree species map produced by the CNN (Figure 4) captured species composition transitions on the elevation transect traversed in the image strip, from a mixed-species composition with oak, cedar, and pine at lower elevation, to fir-dominated forests at higher elevation (Figure 5). We determined the relative proportion of each species based on the number of pixels for each species relative to the total number of pixels at each elevation band. Black oak and incense cedar were primarily found at lower elevations while Jeffrey and sugar pine occur throughout the gradient in varying abundances, with a distinct band of lodgepole pine occurring around 2100 m in flat terrain near a variety of lake and wetland systems in the middle of the imagery strip (Figure 4). Both fir species had the greatest relative abundance at mid- to high elevation, and Jeffrey pine showed the greatest relative abundance at high elevation. However, dead trees were relatively abundant in the highest elevation zone as well, and many of these were probably red and white fir (personal observation). Almost a quarter of the trees identified in the image strip were dead, but the greatest relative abundance of dead trees was at high elevation (2620–2800 m). The general distribution of tree species on an elevation

gradient in Sierran mixed conifer forest has been described [76,77], but this can vary at fine scales locally with variations in local environmental conditions and forest history. Accurate, high-resolution mapping such as has been demonstrated in this study provides spatially-explicit information on these distributions locally and at fine scales.

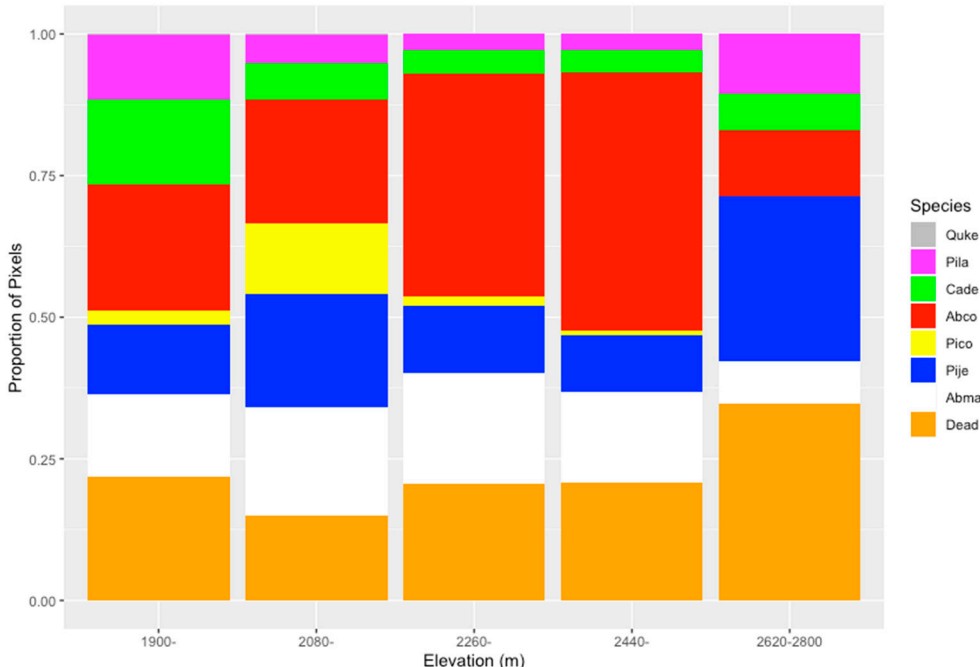

**Figure 5.** The proportion of pixels classified as each species within elevation bands along the elevation gradient for the seven mapped species and dead canopies.

## 4. Discussion

### 4.1. CNN Performance Using Hyperspectral Versus RGB Data

The hyperspectral CNN model outperformed the RGB CNN model by 23% on average, across all metrics and species. These results indicate that the additional spectral information available in the hyperspectral imagery is informative to the classification models and increases model performance. The advantages of spectrally 'deep' hyperspectral imagery over broadband RGB pseudo true color imagery has been demonstrated using discriminant analysis in controlled laboratory settings to monitor the evolution of apples in storage with high (95.83%) accuracy [78]. Similarly, hyperspectral imagery provided more spectral information which aided in the classification of crop plants and weeds in an agricultural setting using statistical discriminant analysis and selected wavelet coefficients to classify five species and weeds at an acceptable (75%–80%) success rate [79]. Hyperspectral imaging proved effective in isolating spectral band combinations to detect *Fusarium* fungal infections in wheat plants in a laboratory setting. These spectral band combinations were not available in broadband RGB imagery [80].

Our method uses additional spectral data from hyperspectral imagery, informing the CNNs models in ways that are unavailable in broadband spectral imagery and at a spatial scale that is appropriate for identifying individual trees to species. CNN models show potential for automatic parameter tuning with a Multi-Layer Perceptron to conduct classification using additional hyperspectral band data which provides valuable information towards material and object recognition which encode pixels spectral and spatial information [81]. To deal with the high dimensionality of the hyperspectral data, we applied a PCA dimensionality reduction, as is standard practice. We also found that L2 regularization helped cope with the relatively small amount of training data available to the model.

We did not include pooling layers in our network architecture to avoid sub-sampling of the input. However, with higher spatial resolution input, we might need pooling in order to avoid having an excessively deep network. To process higher spatial resolution input, we could adopt the standard U-Net approach [82], which combines pooling layers with up-sampling layers, although this would necessitate training on tiles rather than individual patches. Alternatively, we could use dilated convolutions so that our patch-based training procedure is unchanged [83].

## 4.2. CNNs Versus Other Machine Learning Methods for Tree Species Identification

We achieved per-species tree identification with F-scores ranging from 0.67–0.95 using hyperspectral data and relatively simple CNN models and minimal parameter tuning in a mixed-conifer forest (Objective 2). Accurate, automatic species identification of individual trees over large areas give promise that ecological information derived from these maps (e.g., Figure 5) can be used confidently by forest ecologists and resource managers (Objective 3). Our results were achieved under ideal circumstances, contained in a single strip of imagery with a uniform solar illumination angle, using isolated, tall canopies in a temperate forest ecosystem with seven species. Atmospheric correction radiative transfer models can be applied to help mitigate issues caused by multi-date imagery collection for the purpose of species identification [7]. Other efforts to classify individual tree species from hyperspectral imagery using machine learning have yielded accuracies that vary widely depending on the forest type, the remote sensing platform, and the classification methods used. A study also in the Sierra Nevada Mountains used an artificial neural network classifier to identify six conifers species (based on 398 trees) resulting in classification accuracies ranging from 52% for spectral band subsets to 91% accuracy for only 'well lit' canopies [84]. Our study improves on these accuracies by adding nearly double the training samples, using a more flexible neural network architecture, and working in a variety of illumination conditions.

Leveraging co-registered LiDAR and Hyperspectral Imaging and several machine learning classifications approaches including Random Forest, other neural networks, CN2 rules, and Support Vector Machines, previous studies have also addressed tree species identification in a range of forest types (including agroforestry, plantations, and urban forests) worldwide. For example, in natural forests (not plantations) per-species accuracies ranged from 59%–67% in North American deciduous forest [23], but were much higher in a study that classified 14 temperate forest species in Austria with 79%–100% per-species accuracy (average 92%) [8], and for seven species (95%) in mixed evergreen forest in coastal California [9]. Studies in tropical natural forests achieved 85% average accuracy for eight species in dry forest [4], 86% for five species in subtropical forest [37], and 73% for 17 species in wet forest [5]. Higher classification accuracies have been reported for studies in industrial forestry (plantation) settings in Europe, for example, 95% for four species [10,34], and 90% for three species [19]. Our results were comparable with these other machine learning based classification studies in natural forests, but CNN may be more appropriate to the use case because tree species identification is highly contextual in terms of both space and spectra and warrants further testing.

## 4.3. The Potential Uses of High-Resolution Tree Species Maps

Accurate, high-resolution tree species maps are useful in a variety of contexts including estimating drought stress, canopy water content [85], forest inventory, selective logging [86], fire disturbance, succession modeling [87], and estimating biological diversity [88]. In our study, we did not include a large enough geographic area to make large-scale ecological or management inferences from our results, but emergent patterns are apparent. Our study flightline intersects a complex of small lakes and wetlands, surrounded by dense stands of lodgepole pines, a species that is much sparser or absent in the rest of the elevation gradient. These stands of lodgepole pines are well mapped by the hyperspectral CNN model (Figures 4 and 5). In general, forest composition is dominated by firs, particularly at high elevation, but large stands of Jeffrey pine are visible, particularly at mid elevation

(Figure 4). Red and white firs were sometimes confused with each other, which corresponds with anecdotal accounts from field teams, who occasionally confuse the two species on the ground.

Large numbers of standing dead trees are present across the elevation gradient and constitute a larger proportion of total trees at high elevation. Future field collections should include flexible input categories, when species identification might not be 100% certain or when field biologists can identify a dead and decaying tree to a genus level. Such field classifications could help build genus level mortality estimation maps which would be valuable to forest managers. As forest mortality increases in California and elsewhere in part as a result of climate-change-driven "hot drought" [89], maps of tree mortality [83] are critical for forest management and restoration.

### 4.4. Challenges in Computation and Upscaling to Broader Geographic Areas

We have demonstrated that a CNN classifier can be applied to a single flight line of high spatial resolution hyperspectral airborne imagery to identify tree species with high accuracy based on the reflectance spectra of pixels making up tree crowns. This proof-of-concept was carried out for a site in the NEON network located in temperate mixed-conifer forest in western North America, although the methods are applicable in other forest types. Further comparisons are needed to test our methods in more diverse and dense forest ecosystems and under a wider range of illumination conditions. Early steps towards rigorous in-flight fusion of passive hyperspectral imagery with airborne LiDAR have reduced optical geometric distortions and effects of cast and cloud shadowing effects which will be particularly useful in forested environments [90]. If our methods can be scaled up and applied to larger geographic extents, they can address calls to directly monitor biodiversity and species distributions using remote sensing in an era of global change [91], as well as other forest and ecosystem management applications such as forest mortality monitoring [85,92], especially under climate change [89,93].

Additionally, to cover large areas, multiple flight lines are necessary which are subject to variation in collection times, sun angles, weather conditions and potentially different phenological states of the plants which all increase the complexity of using a hyperspectral imager to map broad spatial areas [94]. One of the greatest challenges to this and any method using airborne imagery collected on different dates and under different illumination conditions is that the spectral conditions change from strip to strip or image or image; thus, models trained in one image may not apply across all imagery, limiting the ability to scale across larger datasets [95]. If training data are available for all flightlines/images these challenges can be overcome using separate classifications. For future efforts where CNN species classifications are attempted, ground training data will be necessary across all flight lines, or more realistically, predictions in overlap regions between flight lines will need to be used to train, test, and validate pixels in adjacent flight lines where no training data is present. Performing this classification of adjacent flight lines where no training data exists is the logical next step for this research. Detecting species across dates using hyperspectral imagery can be improved when a radiative transfer model based on atmospheric compensation is applied, but accounting for atmospheric corrections remains a challenge for species classification across dates and conditions, and having spatially extensive training data is important in such cases [7].

There are other operational challenges to extending our approach beyond a single flight line and to a broader geographic range. Hyperspectral data cubes are voluminous and can be noisy. Solutions include dimensionality reduction [96] and 'denoising' using principal component analysis and wavelet shrinkage [97]. Our model is a pixel-based model and aside from the manual canopy delineation process, three-dimensional forest structure was not used for our analysis. Numerous canopy segmentation methods have been developed [98–100] and they could be applied to further the goal of identifying individual trees in a forest, but our method focuses on the imagery as the sole source to inform the classification.

The relatively high CNN model accuracy underscores the importance of the field dataset in training, testing and validation of the prediction model. If a species is included or excluded from the initial training set or if the range of geographic conditions in the image are not sampled, they cannot be



reasonably expected to be included in the predictions. For this reason, a priori knowledge of the forest species composition should be used to guide how many of each target species needs to be sampled as the percentage of input samples should approximate the composition of the forest [101]. Inadequately sampling enough individuals of a certain species will result in under-classification (omission errors) by the CNN. Using CNNs for species classification is a promising approach because of the contextual information used to classify each tree species and often not utilized using other machine learning methods. As computational costs reduce over time, the use of CNNs for this application may become more attractive compared to other machine learning classifiers.

## 5. Conclusions

Our study evaluates Deep Learning CNN models applied to high-resolution hyperspectral and RGB imagery labeled using high precision field training data to predict individual tree species at a pixel level in a natural forest along an elevational and species composition gradient. We present a framework for applying the methods to tree canopies in different ecosystems with similar remote sensing and field datasets. Average classification F-scores for all species was 0.87 for the hyperspectral CNN model and 0.64 for the RGB model. Accuracy results compare quite favorably with previous efforts applying deep learning to image classification and tree species identification, and as we expected, the availability of high resolution hyperspectral image data improved classification accuracy in comparison to broad-band RGB imagery. Our study shows the CNN classifier to be a robust approach to species level classification in Sierran mixed-conifer forest, and points to specific limitations which impact results, such as inaccuracies in canopy segmentation, crown overlap, and similar spectral characteristics of species in the same genus. The methodological framework and CNN code resources are made available for other researchers and forest ecologists to test, repeat, and improve our methods in different forest ecosystems and with different data inputs. We also encourage the development of the code to apply to different image classification problems outside the tree species identification problem, using different training and geospatial imagery sources.

**Author Contributions:** For this research the following represents author contributions to this manuscript. Conceptualization, G.A.F., J.A.W. and J.F.; methodology, G.A.F., J.A.W., J.D.V. and J.F.; software, J.A.W. and J.D.V.; validation, G.A.F., J.A.W. and J.D.V.; formal analysis, G.A.F., J.A.W., J.D.V. and J.F.; investigation, G.A.F., J.A.W., J.D.V., M.P.N., F.W.D. and J.F.; resources, M.P.N., F.W.D. and J.F.; data curation, G.A.F., J.A.W., M.P.N. and J.F.; writing—original draft preparation, G.A.F. and J.F.; writing—review and editing, G.A.F., J.A.W., J.D.V., M.P.N., F.W.D. and J.F.; visualization, G.A.F., J.A.W., J.D.V. and J.F.; supervision, G.A.F., F.W.D. and J.F.; project administration, F.W.D. and J.F.; funding acquisition, F.W.D. and J.F.

**Funding:** Funding was provided by the U.S. National Science Foundation (EF-1065864, -1550653, -1065826 and -1550640) and the Joint Fire Sciences Program (15-1-07-6).

**Acknowledgments:** We thank Marissa Godwin, Matt Hurteau, and field crews from the University of New Mexico, who collected the permanent plot-based tree location data in the Teakettle Experimental Forest. Funding was provided by the U.S. National Science Foundation (EF-1065864, -1550653, -1065826 and -1550640) and the Joint Fire Sciences Program (15-1-07-6). The authors would like to thank D. Tazik, J. Musinsky, T. Goulden and N. Leisso from NEON with their support and patience providing guidance in using NEON data and derived products. We thank Isaiah Mack of Eclipse Mapping who was critical in training and setup of EOS GPS equipment.

**Conflicts of Interest:** The authors declare no conflict of interest. J.W. Contributed to this research while employed at Columbia University and in his personal capacity prior to employment by Amazon Corporation. The opinions expressed in this article are the author's own and do not reflect the view of Amazon Corporation.

## Appendix A

**Table A1.** List of Abbreviations used in this manuscript.

| Abbreviation | Name | Description |
|---|---|---|
| AOP | NEON's Airborne Observation Platform | A remote sensing system composed of an orthophoto camera, LiDAR sensor, and hyperspectral imager. |
| CHM | Canopy Height Model | The canopy height model was used to determine individual tree canopies |
| CNN | Convolutional Neural Network | The classification technique used to predict tree species from remote sensing imagery. This is also called 'deep learning' in the text. |
| DEM | Digital Elevation Model | Last return LiDAR derived surface representing the ground surface in our analysis |
| DSM | Digital Surface Model | The First-return LiDAR derived surface representing the tree canopy surface in our analysis |
| LiDAR | Light Detection and Ranging | The three-dimensional (3D) ranging technology used to measure the CHM, DSM, and DEM. |
| NEON | National Ecological Observatory Network | NEON was responsible for collecting the airborne remote sensing data in 2013. |
| SMA | Spectral Mixing Analysis | Classification method used to discriminate from multiple different spectra in pixels, often used with high spectral resolution imagery |

## Appendix B. Instructions on How to Use the 'Tree-Classification' Toolkit

All data used to run the experiment are multiple gigabytes and are hosted online. All NEON data are available and free for use, contact the National Ecological Observatory Network (NEON) to download the imagery data files for site D17 used in this study: https://data.neonscience.org/home. The HDF 5 files can be converted to a geotiff using R code found here:

http://neonscience.github.io/neon-data-institute-2016//R/open-NEON-hdf5-functions/

Automatic tree species classification from remote sensing data

All code used to run the analysis is located in the online repository here:

https://github.com/jonathanventura/canopy

Files needed:

- Hyperspectral Imagery: data/NEON_D17_TEAK_DP1_20170627_181333_reflectance.tif
- Red-Green-Blue Imagery: data/NEON_D17_TEAK_DP1_20170627_181333_RGB_reflectance.tifCanopy height model: data/D17_CHM_all.tif
- Labels shapefile: data/Labels_Trimmed_Selective.shp

All data to replicate our experiment can be found here:

https://zenodo.org/record/3470250#.XZVW7kZKhPY

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
