# Peer review of "A Convolutional Neural Network Classifier Identifies Tree Species in Mixed-Conifer Forest from Hyperspectral Imagery"

_remotesensing, doi:10.3390/rs11192326_

Round 1

Reviewer 1 Report

The paper aims at the identification of trees in mixed-conifer forest using a convolutional neural
network with hyperspectral imagery. The authors compare tree identification using pseudo RGB data
and hyperspectral data with CNN. This work is very interesting and highlights a new way to identify
species using hyperspectral data. Moreover, the provision of codes and data is a good and unnatural
initiative. Nevertheless, some steps are not clearly explained and this article seems not very robust to
criticism. I will try to make objective and constructive comments to improve it
General comments:
Be more coherent with the species names (vernacular names/scientific names/abbreviations).
The accuracy given is not an accuracy but a precision. Use the F-measure for each class.
You CNN is not detailed enough and your methodology too.
Some comments below:
Line 25: delete (approximately 1 x16 km)
Please, replace “hyperspectral imagery” by “imaging spectroscopy”. This notation is more specific.
Line 28: “15 m” convert to pixel
Line 28-29: You mentioned in the abstract “a CNN model accurately classified 95% of all trees in the
training datasets and 100% of all trees in an independent test dataset”. However, I don’t found these
results in the results part. Please clarify. Moreover, it’s not an accuracy but a precision. Use Fmeasure.
Line 30: “true color”. In the article, you never mention the bands used for the true colour. Moreover,
you do not notify that they come from hyperspectral information, I guess.
Line 102: delete “in question”
Line 151-152: it is not exactly a comparison between RGB and imaging spectroscopy. Specify that the
data is from hyperspectral.
Line 175: Don’t use abbreviation “P. contorta”
Line 197 to 204: Please clarify the pre-treatment of hyperspectral data (atmospheric correction,
which software or method?)
Line 203: Field of View. Do not use capital letters
Line 205: Can you present in a table the number of tree for each species used for this study with
dendrometric information?
Line 273: Your discrepancy come from the DSM resolution used for hyperspectral geo referencing.
Can you add its resolution?
Line 276: When was the high-resolution image acquired? Can you be more specific?
The diameters of the different circles are not equal. Can you specify why?
Line 292: This CNN method is well adapted using RGB, so what are the changes you have made?
Moreover, I'm not sure it's well suited for hyperspectral. Hyperspectral information may be a bit wasted, how many bands do you use?

Line 296: Can you specify for each hyperparameters what they affect/control? Do you use a pretrained
neuronal network?
Line 303: Your models are not available yet. Could you make them available for me to test?
Line 320: You do not precise the size of your train test and validation dataset. Do you think that the
selection of your training can have an impact on the classification? Why?
Line 322: Why is the learning rate not similar for hyperspectral and RGB imagery?
Line 324: Is it the best model for each type of data (hyperspectral and RGB) or only one for both? Do
you use the same data for the training set?
Line 345: You never mention the Table 1. Please mention it or delete it.
Line 347: Why do you mention GPS points? Use the acronym for each species. For each table you use
“accuracy”. However, the “accuracy” is not an accuracy. It is the precision of the class. Use the Fmeasure,
which takes into account the recall and precision. Round to 1 decimal. Do not repeat (%)
Line 352: Same comments.
You do not describe briefly your results and you do not compare it. Please add some paragraphs.
Line 363: Aggregation? What do you mean? Why do you add the training and validation set?
Line 366: Can you put the legend species out of the map? Maybe use the Latin acronym
Line 371: Can you add the results for the fine-tuning?
Line 385: It is not a ‘function’, it is a variation. If the distribution of species is well known, I do not
think this graph is suitable. Use a linear regression for each of the elevation gradients comparing the
theoretical proportion and the estimated proportion
Line 398: Can you clarify?
Line 403: Please, clarify the impossible thing
Line 408: Describe your CNN in methodology part. I do not think you use the same CNN model for
pseudo RGB and hyperspectral data.
Line 412: Not exact
Line 415: Refer to https://www.mdpi.com/2072-4292/11/7/789 that puts forward the problems you
mentioned
Line 419: What does ‘n’ refer to? Add it in the Mat&Method section.
Line 423 to 435: Standardize the name of the methods
Line 431 to 435: For your issues about geographic discrepancy refer to
https://ieeexplore.ieee.org/abstract/document/7852477.
Line 441: Do not add ‘true color ‘, which means that it’s from another sensor than a RGB camera.
Line 455: It is a spectral resolution? Please clarify.
Line 470 to 482. You have a good example of the issues in the following paper:
https://www.mdpi.com/2072-4292/11/7/789

Author Response

We thank the reviewer for their thoughtful and constructive comments. We have responded to all reviewers comments in the attached document.  Reviewers comments are listed in black text and our responses are in red text.    

Reviewer 2 Report

The text is well written and clear. The question is of broad interest and the results as reported are compelling. However, the results seem to be miscalculated and not tallied with care or rigor. Unless I am misreading the tables, none of the values in the confusion matrices add up. From a presentation point of view, I have minor concerns with the text explaining the way the model was chosen and proving additional context for readers coming from deep learning for remote sensing.

The most pressing problem is that the confusion matrices seem to be incorrect. Unless I am missing something, many of the ‘% accurate’ totals are just wrong? I also do not accept showing the training, testing and validation together. There is plenty of space to showing the testing and validation. The training data scores are not relevant. Deep neural networks have millions of parameters, the question of model fit is only relevant to generalization. I cannot follow where the authors claim they have 100% accuracy is testing data (in the abstract), but none of the results reflect this, and the actual values are miscalculated. This does not make the paper seem trustworthy. Normally I would recommend the paper be rejected outright, but I’m concerned there is some confusion on my part that would explain this perceived error.

For example, a random row from in each figure.

Table 1. Abies concolor 109 correctly predicted, 10 incorrect is 109/119 = 91.6%, but the final column reads 90.83%?

Table 2. Calocedrus decurrens. The row sum is 1668 + 15 + 54 + 1 + 1 + 1 = 1740, but the listed total is 1858? The accuracy is 1668/1740 = 0.958? The authors must have some explanation since 1668/1858 = 89.77% which is what is written.

And the most egregious, in Table 3. 13/18 Black Oak trees were predicted correctly, but the table lists accuracy as 100%.

A more minor concern is the text surrounding the introduction to the model. I think L99 is the source of my confusion below. The authors are approaching this from the perspective of using deep convolutional features as a learned kernel for pixel classification. Given the unusual nature of the model compared to other works in deep learning for remote sensing, I think this section needs to be expanded to highlight the kind of rationale and mental framework the authors had when they arrived at their approach. It would probably help reduce the questions I’ve written below.

L290: From a computer vision perspective the approach is unusual, and I think the authors could give the reader a bit more detail on how it was selected, what alternatives were rejected and what are the benefits of this design. I’m not suggesting that the model should not be used just because it seems out of step with computer vision practices. Only that the authors should help prepare the readers, who would have arrived at the paper with different expectations. This statement also goes for using the entire hyperspectral cube instead of choosing pertinent bands or trying dimensionality reduction. See

Maschler, J., Atzberger, C., Immitzer, M., 2018. Individual Tree Crown Segmentation and Classification of 13 Tree Species Using Airborne Hyperspectral Data. Remote Sens. 10, 1218. https://doi.org/10.3390/rs10081218

My understanding, I welcome to be corrected, is that the proposed approach is a kind of hybrid between semantic segmentation (e.g Unet https://github.com/reachsumit/deep-unet-for-satellite-image-segmentation) and sliding window classification (https://www.mdpi.com/2072-4292/11/15/1786). The authors state

“We did not use zero padding, and the number of convolutional layers was chosen as (L-1)/2 so that the CNN would output a single prediction for the center pixel in the window under consideration. At test time, the model is run in a fully convolutional manner to produce a prediction at each output pixel in parallel”

This seems clever way to force conv layers to act similar to traditional kernel operations, but also incredibly inefficient. So for a 400 pixel image, the model would be run 400 times? What happens at the edges of the image, i.e when trying to predict an edge pixel as the center of a spatial kernel? Help the reader understand why this strategy was adopted. It seems like a kind of spatial neighborhood function, where the features in the convolutional layers are helping to inform the prediction of the central pixel using the pixel values in the surrounding cells. Can the authors point to any other work that employs this strategy in deep learning?

There are couple other things that are unusual about its architecture compared to the typical computer vision model. Perhaps the authors have reasons for their format compared to more traditional models. If so, it would be useful in stating which other types of architectures were tried and were found to be less accurate. I’m being less critical and more genuinely curious!

Why not use a more standard model architecture such as a UNET? It could be run just once. Alternatively, one could treat each tree as an image and perform classification on each “chip”. The benefits here are avoiding the extra step of majority voting. The downside is some distortion during image resizing when inputting into the model. My experience is that this distortion is relatively minor. This kind of discussion would be really useful in the intro to guide the reader about the authors particular rationale, rather than just presenting the proposed method without context. The lack of pooling layers surprises me. Pooling helps decrease the sensitivity of the model to small perturbations in the input environment. This significantly increases generalization. Again, it may have to do with the center-pixel prediction idea. More text here would help the reader give context.

Very Minor Comments

L75: I’m slightly uncomfortable with the word ‘pioneered’. Asner has done great work, but it feels out of place. Plenty of hard-working people worked on this for many decades!

I’ve gone through the code and it looks reasonable and well commented. Is it currently in a private repo? Why is the github link broken? I would strongly encourage the authors to release the label data, I don’t see the need to hold on to it. It would be helpful in comparing with future methodology and increase the citation value of this work.

For simplicity, L314 means that labeled images (pixel representations) for the same tree does not appear in training and testing? I’m pretty sure that’s what the authors are saying, but I think it is worth being explicit.  

I think the figure legend in Table 3 should mention that this is the majority voting, whereas table 2 is per-pixel? This would help the reader remember quickly. I’m not sure what “the table above is an aggregation of training, validation and test datasets” means. Why not just show validation data accuracy? That’s why it was held back from training and testing.  

Please add the mean accuracy across all classes either in the table or in the results text. Would answer the question, Is RGB or Hyperspectral better on average?

Author Response

(The authors gave the same response as above.)

Reviewer 3 Report

Major comments

The paper explored a CNN architecture to classify seven tree species of a mixed-conifer forest, reaching surprisingly 100% of accuracy. Due the lack of studies using CNN for this purpose until now, it is an interesting topic, however I have some concerns about the study:

-          There are a lot of papers that successful classified tree species using hyperspectral data and machine learning methods. The authors need to justify their choice by CNN instead of such methods. They did not mention the advantages of CNN over algorithms as RF and SVM. Since CNN involves a high computational cost, not always would be the most attractive option. They claim in introduction that there are a lack of methods and data to deal with tree species classification, which is not true.

-          It is not clear how the samples were split in training, validation and testing set. They stated that they used an independent test set, however they must clarify that for ‘independent’ you mean ‘pixels that were not used to train the classifier’ or entire trees that were not used in training step. Each tree must be treated as a basic unit in classification analysis, you cannot used pixels (even being different pixels) but from the same crown to train and test the classifier. It may provide an unrealistic accuracy estimates. They need to write how many pixel and tree samples were used for each tree species/dataset? Were the samples split only one time?

-          As I understood they made four experiments, but in methodology they were not clearly stated. Would be: RGB-polygons, RGB-pixels, Hyperspectral-polygons, hyperspectral-pixel?

-          Result section is very weak: they put the confusion matrices, but they did not present the results. Why some specific species were more confused with other species, etc..? Commission and omission errors? Which species have more increase when you include all hyperspectral dataset instead of RGB bands?

-          The discussion section is long, but it not fully matched with the results. They discuss about topics that they did not explored. They did not perform any specific tests here regarding changing the number of samples, or tree species classes, or comparing with other study area or flight strip. They did not discuss about the commissions and omissions errors of each species that you classify and possible reasons that could led to that (e.g., fewer number of samples). They wrote three paragraphs talking about future works, about the necessity of representative samples, but they did mention it before, even how many samples per species you use, or how it could influence your result: species with less samples would present lower or more variability in accuracy? Would be underpredicted?  Characteristic of species that can lead to more confusions? How CNN can deal with issues, as imbalanced sample set or fewer number of samples?

Specific comments:

The abstract is confused. Until line 30, you presented your data, methods and results, and after that you started again writing about methods. Please, reallocate line 30 to 35 to before the accuracy results.

Line 49 to 50- Careful with this information, it is not lack of operational method. If you look deeply in literature, there were much more studies concerning tree species mapping than your references 1-3. See the review of Fassnacht et al. (2016), and papers below that as your study, used high spectral and spatial data for tree species classification.

Line 59- this sentence refers to your goal, remove it from here.

Line 61- I do not agree with your items 1 and 2: first: there are available high spectral and spatial resolution data to identify trees at the crown level, e.g, hyperspectral cameras onboard of UAV or airplanes can provide it and are extensively used for this purpose. 2) with a proper classification scheme (i.e. representative sample set, parameters optimization, features extraction or selection), machine learning methods are sophisticated enough to deal with the high variability of the dataset. They were successful applied in a lot of studies involving hyperspectral data for tree species classification, please see:

Ferreira, M.P.; Zortea, M.; Zanotta, D. C.; Shimabukuro, Y. E.; Souza Filho, C.R. de. Mapping tree species in tropical seasonal semi-deciduous forests with hyperspectral and multispectral data. Remote Sens. Environ. 2016, 179, 66–78.

Shen, X.; Cao, L. Tree-Species Classification in Subtropical Forests Using Airborne Hyperspectral and LiDAR Data. Remote Sens. 2017, 9, 1180.

Dalponte, M.; Ørka, H.O.; Gobakken, T.; Gianelle, D., Næsset, E. Tree species classification in boreal forests with hyperspectral data. IEEE Trans. Geosci. Remote Sens. 2013, 51, 2632–2645.

Féret, J.; Asner, G. P. Tree species discrimination in tropical forests using airborne imaging spectroscopy. IEEE Transactions on Geoscience and Remote Sensing 2013, 51, 73–84.

Maschler, J.; Atzberger, C.; Immitzer, M. Individual Tree Crown Segmentation and Classification of 13 Tree Species Using Airborne Hyperspectral Data. Remote Sens. 2018, 10, 1218; doi:10.3390/rs10081218.

Nevalainen, O.; Honkavaara, E.; Tuominen, S.; Viljanen, N.; Hakala, T.; Yu, X.; Hyypa, J.; Saari, H.; Polonen, I.; Imai, N.N.; Tomaselli, A.M.G. Individual Tree Detection and Classification with UAV-Based Photogrammetric Point Clouds and Hyperspectral Imaging. Remote Sens. 2017, 9, 185.

Ballanti, L.; Blesius, L.; Hines, E.; Kruse, B. Tree Species Classification Using Hyperspectral Imagery: A Comparison of Two Classifiers. Remote Sens. 2016, 8, 445.

Ghosh, A.; Fassnacht, E. F.; Joshi, P. K.; Koch, B. A framework for mapping tree species combining hyperspectral and LiDAR data: Role of selected classifiers and sensor across three spatial scales. Int. J. Appl. Earth Obs. Geoinf. 2014, 26, 49–63, 2014.

Tuominen, S.; Näsi, R.; Honkavaara, E.; Balazs, A.; Hakala, T.; Viljanen, N.; Pölönen, I.; Saari, H.; Ojanen, H. Assessment of Classifiers and Remote Sensing Features of Hyperspectral Imagery and Stereo-Photogrammetric Point Clouds for Recognition of Tree Species in a Forest Area of High Species Diversity. Remote Sens. 2018, 10, 714

Piiroinen, R.; Heiskanen, J.; Maeda, E.; Viinikka, A.; Pellikka, P. Classification of Tree Species in a Diverse African Agroforestry Landscape Using Imaging Spectroscopy and Laser Scanning. Remote Sens. 2017, 9, 875.

Raczko, E.; Zagajewski, B. Comparison of support vector machine, random forest and neural network classifiers for tree species classification on airborne hyperspectral APEX images. Eur. J. Remote Sens.2017, 50, 144–154.

Dalponte, M.; Ene, L. R.; Marconcini, M.; Gobakken, T.; Næsset, E. Semi-supervised SVM for individual tree crown species classification. ISPRS J. Photogramm. Remote Sens. 2015, 110, 77–87. doi: 10.1016/j.isprsjprs.2015.10.010.

Graves, S.J.; Asner, G.P.; Martin, R.E.; Anderson, C.B.; Colgan, M.S.; Kalantari, L. and Bohlman, S. Tree species abundance predictions in a tropical agricultural landscape with a supervised classification model and imbalanced data. Remote Sens. 2016, 8, 2, 161.doi: 10.3390/rs8020161.

What you can argue as advantage regarding the CNN, is the not needed of feature extraction phase (or as they called handcrafted features), since the algorithm automatic extract deep features.

Line 74- Please clarify, however [21]… ?

In introduction there is a lack of comparison among studies that used machine learning methods: like SVM and RF. These algorithms also reached good results in tree species classifications (papers above). Since the CNN requires much more processing time and computer power, what are the advantage of this method in comparison with machine learning? Since you did not test them in your dataset. In line 101 you put some information that justify your choice (CNN), the use of contextual information, automatic feature extraction… but you need to link it with machine learning methods, clarify the advantage of your method. And also, if the method is justified due to the high computational cost.

Line 109- this sentence is long without comma, point, please rewrite.

Line 116- (Yann et al. 2015) must be enumerated

Line 123- this sentence is very confused, please rewrite: the need to program the code needed to apply…

Figure 1- What does that mean the abbreviations in Field GPS species, please specify in figure caption.

Line 161- You start writing about the data collection, then in line 197 you back to data collection. Please organized the topic in study area description and data description, not mixing them.

Line 239- What do you mean with: total of 920 canopies were originally recorded, 920 tree crowns?

Line 276- How did you decide the circle sizes? According to the tree crown sizes?

Item 2.4- Details about CNN architecture are missing: number of kernels for each convolutional layer? What was the optimization function? Window size, was 3x3? Pooling layers? What was the proportion of training, validation and test samples? How were they separated?

Actually, I saw that only in discussion you put that you did not use pooling layer. Please detail it here in methods. Why did you opted for this and that decision imply in what regarding the computational cost? Since pooling layers are used to reduce the spatial size of feature maps significantly and, consequently, the computation volume for the next layers to be processed. If in further studies you will deal with more flight strips, the same decision can be made?

About the samples, you must clarify how many trees (and pixels) were used in each sample set (training, validation, test), and how did you split them? Did you split them only one time?

Item 2.5- I don’t understand how the training and test set was separated. You stated that you used an independent test set, however you must clarify that for ‘independent’ you mean ‘pixels that were not used to train the classifier’ or entire trees that were not used in training step. Each tree must be treated as a basic unit in classification analysis, you cannot used pixels (even being different pixels) but from the same crown to train and test the classifier. In fact, you result of 100% of accuracy makes me suspicious that only the pixels, but not the trees, were separated in training and test set, please clarify. You can find more information about the importance in keep the individual tree crown as classification unit in Baldeck and Asner (2014):

“When spectral differences exist among crowns, then classification accuracies that are evaluated at the pixel level—with pixels randomly assigned to training and test datasets regardless of crown membership—may provide unrealistic accuracy estimates.”

Baldeck, C. A., & Asner, G. P. (2014). Improving remote species identification through efficient training data collection. Remote Sensing, 6, 2682–2698.

Line 326- How the accuracy was evaluated? Did you do the inference over the image patches?

Line 336- comparing the resulted accuracy…

Please rewrite- The best hyperparameters settings were…

Line 340- Were the same CNN parameters applied to RGB and hyperspectral data?

Table 1. You did not call Table 1 in the text. Please check.

Table 2. Please check the caption: it would be the table below, not above. Also correct:

The table below corresponds to the pixel CNN result in which each pixel is treated individually

Please, call the tables before they appear in the text. The confusion matrices order is confused: why did you first put the confusion matrix of polygons corresponding to RGB classification, then for pixel classification using hyperspectral, and then hyperspectral aggregation of training, validation and test datasets?

Please, follow a logic order: first presenting only polygon and pixel results of RGB, after hyperspectral or vice-versa.

Table 3 is confuse, why did you aggregate training, validation and test dataset to build the confusion matrix? Or did you mean the validation regarding polygons instead of pixels? You must put only regarding the test set, otherwise it doesn’t make sense.

Again, correct: the table above…

In results section you did not make it clear how much increase in accuracy you reached when using hyperspectral data or RGB data. You put the confusion matrices, barely talking about your results. Also, regarding the confusion errors among species. Why some specific species were more confused with other species, etc..? Commission and omission errors? Which species have more increase when you include all hyperspectral dataset instead of RGB bands? Besides present the confusion matrices you must highlighted your result. A chart showing the overall accuracies of each classification would be interesting (RGB-pixel, RGB-polygons, hyperspectral-pixel, hyperspectral-polygons).

Figure 5. Put the entire word for elevation and prop. What does that mean? proportion? How it was calculated?

Lines 391 to 393 refers to which reference? Which baselines classification techniques?

Line 395- the reference must be enumerated Long et al. (2015)

Line 397- what did you mean with ‘further machinery’?

Line 400- here you mentioned that you did not use pooling layer, you suppose to write this in methodology section too as my comment above.

Line 404- This sentence is very confused, please rewrite. Our method uses additional spectral data from hyperspectral imagery, informing the CNNs models in ways that are impossible in broadband spectral imagery and at a spatial scale that is appropriate for identifying individual trees to species.

Which additional spectral data? What did you mean with: informing the CNNs models in ways that are impossible in broadband spectral imagery?

Line 407: it doesn’t make sense: To deal with the high dimensionality of the hyperspectral data, we used a large number of features in each convolution layer.

I mean, here you are saying that to deal with the high dimensionality you used a large number of features? This would increase the dimensionality.

Line 422- and working in a variety of illumination conditions? But you mentioned above that you have the same illumination conditions.

Lines 436- 449- You put all this paragraph talking the advantages of hyperspectral data, but in result you did not clarify how much accurate for each species was the use of hyperspectral in comparison with RGB results. Also, you mentioned some studies using low spatial resolution data, which is not your case. There are in literature many works focusing on tree species classification using high resolution hyperspectral data, that would be a fairer comparison (see the references that I put above). If you have this kind of data you certainly would not use SMA, that is required for lower spatial resolution.

Line 454-456- As I said, you must put more about it in your results.

Lines 483-498- I agree with your considerations here, but I don’t feel that they match with your study. You did not perform any specific tests here regarding changing the number of samples, or tree species classes, or comparing with other study area or flight strip. You did not discuss about the commissions and omissions errors of each species that you classify and possible reasons that could led to that (e.g., fewer number of samples).

Line 502- your repeat this information in line 510: We present a framework for applying the methods to tree canopies in different ecosystems with similar remote sensing and field datasets.

Line 504- You put it in conclusion but you did not discuss about it previously: Misclassifications occurred, particularly among the black oak, which were smaller and less numerous than the larger conifer trees in our study area

In general, I feel like the results did not fully match with your discussion. You did not make it clear regarding what do you want to prove with your methodology. Since you barely put the confusion matrices and a figure with the proportion of each species according to elevation level. Your discussion is long, but with information not related with you explore here: you attained three paragraphs talking about future works, you talk about the necessity of representative samples, but you did mentioned it before, even how many samples per species you use, or how it could influence your result: species with less samples would present lower or more variability in accuracy? Would be underpredicted?  How CNN can deal with that? Even with lower number of samples, CNN provide some methods for data augmentation.

And also, there was a lack of comparisons with the most used methods for tree species classification using high spatial and spectral data (like SVM and RF), as your case. Not comparing your study with studies using lower spatial resolution data.

Author Response

(The authors gave the same response as above.)

Round 2

Reviewer 2 Report

I am reviewer 2 from the first revision. The authors have met most of my concerns and helped improve the clarity of the paper. Especially in clarifying the tables. I would encourage the authors to make the github link public and place the training locations there (not the hyperspec tiles). The RS data can be redownloaded from NEON easily. The paper will get cited more if the data are included.

Author Response

Response:  We have updated our github link to a public bitbucket link with the training locations here: https://bitbucket.org/jonathanventura/tree-classification/src/master/

We appreciate the reviewer’s constructive comments. 

Reviewer 3 Report

Dear authors,

Thanks for improving the paper. I have few considerations below, mainly regarding the structure of the topics.

I could not open the repository with the code, please check.

Besides, I still suggest a review in English writing before the publication.

line 26- ..also used..

line 35- what do you mean with 'combined' F-score? average its ok. You can remove the information about precision and recall from the abstract.

line 76- training the classifiers..

line 82- it is not necessary to be airborne, currently, UAV sensors also provide data with both high spectral and spatial resolution and have been used for tree species classification.

Nevalainen, O.; Honkavaara, E.; Tuominen, S.; Viljanen, N.; Hakala, T.; Yu, X.; Hyypa, J.; Saari, H.; Polonen, I.; Imai, N.N.; et al. Individual Tree Detection and Classification with UAV-Based Photogrammetric Point Clouds and Hyperspectral Imaging. Remote Sens. 20179, 185.

Tuominen, S.; Näsi, R.; Honkavaara, E.; Balazs, A.; Hakala, T.; Viljanen, N.; Pölönen, I.; Saari, H.; Ojanen, H. Assessment of Classifiers and Remote Sensing Features of Hyperspectral Imagery and Stereo-Photogrammetric Point Clouds for Recognition of Tree Species in a Forest Area of High Species Diversity. Remote Sens. 201810, 714.

Sothe, C.; Dalponte, M.; Almeida, C. M. De; Schimalski, M. B.; Lima, C. L.; Liesenberg, V.; Miyoshi, G. T.; Tommaselli, A. M. G. Tree Species Classification in a Highly Diverse Subtropical Forest Integrating UAV-Based Photogrammetric Point Cloud and Hyperspectral Data. Remote Sens. 2019, 11, 11, 1338.

Line 378- 384: Is this the same scheme that you mention in line 401? Because if it is, there is a confusion between the terms ‘testing’ and ‘validation’. Please check:

https://towardsdatascience.com/train-validation-and-test-sets-72cb40cba9e7

Also the paragraph seems lost here, since the title is ‘Data preparation’ and you expect something about hyperspectral data and then you started writing about the evaluation method.

Lines 420- 434. This part can be placed together with lines 378-384 as an ‘accuracy assessment’ topic; and lines 385 to 419 together in the topic ‘Optimization / Hyperparameter Tuning / Prediction’ before the accuracy assessment.

Line 442- …because the proportion of samples per class is imbalanced…

Line 446-447- You already said it before.

Lines 438-452 can be placed in methodology section; they do not present a result. The results started in 452 The CNN classifier using…

Line 510- Please, rewrite: in which 16/47 of the trees in the sample were classified as a white fir, and only 14 trees 510 were correctly classified (Table 4). ..White fir (standardize the species names regarding the use of upper or lower case).

Line 514: .. 16 Jeffrey pines trees..

Line 559: CNN performance using hyperspectral versus RGB data

Line 678: consider for future works to not label a tree species sample if exist any doubt regarding the species class.

Author Response

Dear authors,

Thanks for improving the paper. I have few considerations below, mainly regarding the structure of the topics.

I could not open the repository with the code, please check.

Response:  We have updated our github link to a public bitbucket link with the training locations here: https://bitbucket.org/jonathanventura/tree-classification/src/master/

We appreciate the reviewer’s constructive comments and we will host our training samples on Zenodo.org for increased repeatability. 

Besides, I still suggest a review in English writing before the publication.

Response:  We have conducted a review for English writing.

line 26- ..also used..

Response:  Corrected.

line 35- what do you mean with 'combined' F-score? average its ok. You can remove the information about precision and recall from the abstract.

Response:  Corrected and removed precision and recall information from the abstract. 

line 76- training the classifiers..

Response:  Corrected

line 82- it is not necessary to be airborne, currently, UAV sensors also provide data with both high spectral and spatial resolution and have been used for tree species classification.

Nevalainen, O.; Honkavaara, E.; Tuominen, S.; Viljanen, N.; Hakala, T.; Yu, X.; Hyypa, J.; Saari, H.; Polonen, I.; Imai, N.N.; et al. Individual Tree Detection and Classification with UAV-Based Photogrammetric Point Clouds and Hyperspectral Imaging. Remote Sens. 2017, 9, 185.

Tuominen, S.; Näsi, R.; Honkavaara, E.; Balazs, A.; Hakala, T.; Viljanen, N.; Pölönen, I.; Saari, H.; Ojanen, H. Assessment of Classifiers and Remote Sensing Features of Hyperspectral Imagery and Stereo-Photogrammetric Point Clouds for Recognition of Tree Species in a Forest Area of High Species Diversity. Remote Sens. 2018, 10, 714.

Sothe, C.; Dalponte, M.; Almeida, C. M. De; Schimalski, M. B.; Lima, C. L.; Liesenberg, V.; Miyoshi, G. T.; Tommaselli, A. M. G. Tree Species Classification in a Highly Diverse Subtropical Forest Integrating UAV-Based Photogrammetric Point Cloud and Hyperspectral Data. Remote Sens. 2019, 11, 11, 1338.

Response:  UAVs are also airborne, they’re just not “manned” (piloted by an on-board pilot).  It is therefore accurate to use the terminology ‘airborne’ to encompass UAVs and manned airborne imagery collections.  We added a sentence at the end of the paragraph which cites these papers that clarifies this point by explicitly distinguishing manned versus UAVs.

Line 378- 384: Is this the same scheme that you mention in line 401? Because if it is, there is a confusion between the terms ‘testing’ and ‘validation’. Please check:

https://towardsdatascience.com/train-validation-and-test-sets-72cb40cba9e7

Also the paragraph seems lost here, since the title is ‘Data preparation’ and you expect something about hyperspectral data and then you started writing about the evaluation method.

Response:  We clarified the text and moved the paragraph as described in the following response.

Lines 420- 434. This part can be placed together with lines 378-384 as an ‘accuracy assessment’ topic; and lines 385 to 419 together in the topic ‘Optimization / Hyperparameter Tuning / Prediction’ before the accuracy assessment.

Response:  Corrected. We moved the paragraph 420-434 to Section 2.5 Data Preparation after the sentences between 378-384, and moved the last paragraph to the topic of ‘Optimization/Hyperparameter Tuning/Prediction before the accuracy assessment. 

We also added a sentence about data augmentation and fixed a table caption.

Line 442- …because the proportion of samples per class is imbalanced…

Response:  Corrected.  Added text to the end of the paragraph to qualify the statement. 

Line 446-447- You already said it before.

Response:  Corrected. We removed this sentence. 

Lines 438-452 can be placed in methodology section; they do not present a result. The results started in 452 The CNN classifier using…

Response:  We agree that this section does contain methods and results because while we do describe the hyper parameter tuning methods, we also report the best hyper parameters which are results of the tuning process. We moved the first sentence to Methods and kept the results sentences in Results

Line 510- Please, rewrite: in which 16/47 of the trees in the sample were classified as a white fir, and only 14 trees 510 were correctly classified (Table 4). ..White fir (standardize the species names regarding the use of upper or lower case).

Response:  Changed all species names to be lower case in the text and capitalized first letter in the table or when at beginning of a sentence. The exception is Jeffrey pine because Jeffrey is a proper name and is always capitalized.

Line 514: .. 16 Jeffrey pines trees..

Response:  Corrected.

Line 559: CNN performance using hyperspectral versus RGB data

Response:  Corrected, we used the reviewers wording for the section title

Line 678: consider for future works to not label a tree species sample if exist any doubt regarding the species class.

Response:  This is an interesting suggestion and a good idea.  We added additional sentences in Section 4.3 to suggest future field collection strategies.